# Differentiation Capacity of Bone Marrow-Derived Rat Mesenchymal Stem Cells from DsRed and Cre Transgenic Cre/*loxP* Models

**DOI:** 10.3390/cells11172769

**Published:** 2022-09-05

**Authors:** Hsiang-Ching Tseng, Menq-Rong Wu, Chia-Hsun Lee, Jong-Kai Hsiao

**Affiliations:** 1Department of Medical Imaging, Taipei Tzu Chi General Hospital, Buddhist Tzu-Chi Medical Foundation, New Taipei City 23142, Taiwan; 2Department of Research, Taipei Tzu Chi General Hospital, Buddhist Tzu-Chi Medical Foundation, New Taipei City 23142, Taiwan; 3School of Medicine, Tzu Chi University, Hualien 97004, Taiwan

**Keywords:** differentiation, Cre and *loxP* system, mesenchymal stem cells, cell fusion, stemness, proliferation, cell cycle progression

## Abstract

Cre/*loxP* recombination is a well-established technique increasingly used for modifying DNA both in vitro and in vivo. Nucleotide alterations can be edited in the genomes of mammalian cells, and genetic switches can be designed to target the expression or excision of a gene in any tissue at any time in animal models. In this study, we propose a system which worked via the Cre/*loxP* switch gene and DsRed/emGFP dual-color fluorescence imaging. Mesenchymal stem cells (MSCs) can be used to regenerate damaged tissue because of their differentiation capacity. Although previous studies have presented evidence of fusion of transplanted MSCs with recipient cells, the possibility of fusion in such cases remains debated. Moreover, the effects and biological implications of the fusion of MSCs at the tissue and organ level have not yet been elucidated. Thus, the method for determining this issue is significant and the models we proposed can illustrate the question. However, the transgenic rats exhibited growth slower than that of wild-type rats over several weeks. The effects on the stemness, proliferation, cell cycle, and differentiation ability of bone marrow–derived rat MSCs (BM-rMSCs) from the models were examined to ensure our design was appropriate for the in vivo application. We demonstrated that MSC surface markers were maintained in DsRed and Cre transgenic rMSCs (DsRed-rMSCs and Cre-rMSCs, respectively). A WST-8 assay revealed decreased proliferative activity in these DsRed-rMSCs and Cre-rMSCs; this result was validated through cell counting. Furthermore, cell cycle analysis indicated a decrease in the proportion of G1-phase cells and a concomitant increase in the proportion of S-phase cells. The levels of cell cycle–related proteins also decreased in the DsRed-rMSCs and Cre-rMSCs, implying decelerated phase transition. However, the BM-rMSCs collected from the transgenic rats did not exhibit altered adipogenesis, osteogenesis, or chondrogenesis. The specific markers of these types of differentiation were upregulated after induction. Therefore, BM-rMSCs from DsRed and Cre transgenic models can be used to investigate the behavior of MSCs and related mechanisms. Such application may further the development of stem cell therapy for tissue damage and other diseases.

## 1. Introduction

Mesenchymal stem cells (MSCs) are multipotent cells capable of self-renewal, and they can give rise to tissue-specific cell types [1,2,3]. Owing to MSCs having been found as a potential source for distinct cells after differentiation, many treatments are being carried out in animal experiments and clinical trials [4,5,6,7,8]. Cell fusion is regarded as a powerful attempt toward tissue regeneration. It repeatedly happens during organ development and in adults [9,10,11]. Additionally, cell fusion after stem cell treatment has been discovered in neurons, pancreatic cells, and other tissues [12,13,14,15,16]. However, the significance of cell fusion has not been fully explored, and the possibility of MSC transdifferentiation after stem cell therapy in damaged tissues remains uncertain [17,18,19]. To date, most of the methods for the in vivo assessment of fusion rely on tissue procurement and histological analysis. Therefore, an in vivo trafficking system should be developed. Molecular imaging can be used to accurately detect fusion in vivo over time and enable the trafficking of MSCs and in vivo monitoring of molecular interactions, which are useful practices in regenerative medicine [20,21].

To detect the fusion of MSCs after their transplantation into organs, a Cre/*loxP*-based model was proposed in this study; the concept of the model is shown in Appendix A. A gene cassette was used to encode a green fluorescent protein gene adjacent to a floxed segment of the *DsRed* gene (which encodes red fluorescent protein) with a stop codon. When cells expressing this cassette fuse with cells expressing the Cre recombinase, cleaving occurs at the *loxP* site, removing the *DsRed* gene and allowing the initiation of transcription of the green fluorescent gene.

Hamilton and Abremski discovered a site-specific recombinase, Cre recombinase, in the P1 phage in 1984 [22]. This recombinase identifies gene sequences by using specific permutations of the *loxP* identification sequence to catalyze genetic recombination. The Cre recombinase and *loxP* sequence are both necessary for the initiation of the recombination reaction. Hence, this system can be applied in different ways such as fate mapping and gene regulation [23,24,25,26]. It was well known that *Cre* and *loxP* have a unique role in DNA recombination. However, there have some concerns about the use of this system. Several studies indicated that the Cre recombinase expression has an impact on DNA damage, growth inhibition, and cell death in mammalian cells [27,28,29]. Since the *Cre* gene and *loxP* sequence do not occur naturally in MSCs, the effects of the *Cre* gene and the *loxP* sequence in MSCs need to be investigated.

In the present study, a growth-inhibitory effect was observed in the transgenic rats compared with the wild-type rats, prompting us to understand if the in vivo application of our design was suitable and whether the characteristic of stem cells was not altered by transgenes. Therefore, we aimed to examine stemness, cell proliferation, cell cycle progression, and cell differentiation of BM-rMSCs obtained from DsRed and Cre transgenic models.

## 2. Materials and Methods

### 2.1. Animal Care and Protocol Approval

Two transgenic rat models, SD-Tg (UBC-DsRedT3-emGFP)18Narl and SD-Tg (UBC-Cre/ERT2)7Narl, were created using Sprague–Dawley rats purchased from the National Laboratory Animal Center (Taipei, Taiwan). The protocol followed for breeding and care was reviewed and approved by the Institutional Animal Care and Use Committee of Taipei Tzu Chi General Hospital, Buddhist Tzu Chi Medical Foundation (102-IACUC-024). The rats were kept under conditions recommended in the National Institutes of Health Guidelines on the Use of Laboratory Animals.

### 2.2. Collecting and Cultivation of Bone Marrow–Derived Mesenchymal Stem Cells (MSCs)

Both femurs and tibias were obtained from male Sprague–Dawley rats aged 6–8 weeks. The bones were imbedded in 75% alcohol for 1–2 min and immersed in a phosphate-buffered saline (PBS) containing 100 µg/mL streptomycin and 100 U/mL penicillin; this was performed in triplicate. The bones were them transferred to high-glucose Dulbecco’s modified Eagle’s medium (DMEM; Gibco, BRL, Grand Island, NY, USA) with 10% fetal bovine serum (FBS) supplement (HyClone, Logan, UT, USA), and adding 100 µg/mL streptomycin, 100 U/mL penicillin, and 4 mM L-glutamine (Gibco). For bone marrow collection, the bone marrow cavity was carefully exposed and injected with a culture medium through a 23-guage needle. The bone marrow tissue was centrifuged at 1500 rpm for 5 min after the addition of Ficoll–Paque PREMIUM (GE Healthcare, Munich, Germany). The supernatant was removed and centrifuged at 2400 rpm for 20 min, and the cell layer was harvested and resuspended in DMEM. The cell-containing medium was centrifuged again at 1500 rpm for 5 min and plated in the culture medium containing DMEM and antibiotics. After 24 h, the attached cells were collected and plated in a culture dish with fresh culture medium. We changed the culture medium every 2–3 days, and the plated cells were passaged when the confluence reached 80%.

### 2.3. MSCs Transfection and Transduction

Bone marrow–derived rat MSCs (BM-rMSCs) were plated in a culture dish supplemented with the aforementioned culture medium. Lentiviral constructs were used to label the BM-rMSCs and served as experimental controls. The lentiviral vector pLAS5w.PtRFP-I2-Puro was procured from the RNAi Consortium at Academia Sinica. Cells (293T cells; American Type Culture Collection, CRL-11268) grown in a culture medium were infected with viral particles; 5 × 10^4^ BM-rMSCs were seeded in 35 mm well plates and left for 24 h. The cells were transduced at a multiplicity of infection (MOI) of 50. Viral production was executed according to the procedure specified in Nature Protocols [30]. After viral packaging and infection, all the cells were placed in a culture medium containing 1 µg/mL puromycin (Sigma, St. Louis, MO, USA) for at least 10 days. The expression of RFP was revealed through flow cytometry and Western blotting.

### 2.4. Transgene Expression of Transgenic Rats

The transgenic rats were killed through CO_2_ asphyxiation, and the transgene expression of the brain, spleen, liver, kidney, and BM-rMSCs was examined through reverse transcription polymerase chain reaction (RT-PCR), immunohistochemistry (IHC), or Western blotting. The experimental procedures are detailed in the following sections.

### 2.5. Phenotypic Characterization

To determine whether the BM-rMSCs maintained their phenotypes, growth after three passages was measured through immunofluorescence staining. The BM-rMSCs were plated on a Lab-Tek eight-well chamber slide (Thermo Fisher, Ottawa, Canada) for 24 h and maintained in a culture medium at a concentration of 2 × 10^3^ cells/well. The cells were then washed with PBS before being added to a solution of 4% paraformaldehyde (Sigma-Aldrich, St. Louis, MO, USA) and PBS for 10 min. Nonspecific binding was blocked using a 1% bovine serum albumin (BSA) solution for 1 h at room temperature. The BM-rMSCs were incubated overnight at 4 °C with four surface markers: CD29 (integrin b1 chain 1:200), CD90 (1:200), CD45 (1:200), and CD11b/c (1:200; all from Biolegend, San Diego, CA, USA The cells were washed and incubated with either fluorescein isothiocyanate (FITC)-conjugated secondary anti-rabbit or anti-mouse IgG antibodies (Thermo Fisher Scientific) at room temperature for 90 min. Subsequently, the slides were washed by PBS and stained with a DNA-binding dye, 4′,6-diamidino-2-phenylindole (DAPI; 5 µg/mL; Invitrogen, Waltham, MA, USA), in PBS for 10 min at room temperature. The cell imaging was captured using an inverted microscope (Eclipse TS100; Nikon, Tokyo, Japan). In addition, flow cytometry was performed to detect the cell surface markers. The cells were plated at 200 cells/cm^2^ on fibronectin-coated culture dishes and harvested at approximately 70% confluence. Thereafter, 10^5^–10^6^ cells were incubated with FITC-conjugated anti-CD29, anti-CD90, anti-CD45, and anti-CD11b/c antibodies for 30 min at 4 °C under dark conditions. The selected antibodies were used at 1:200 dilution. Isotype-matched irrelevant polyclonal antibodies were used as negative controls. The cells were washed and resuspended in cell staining buffer (Biolegend, San Diego, CA, USA) and then analyzed using a flow cytometer (BD Biosciences, San Jose, CA, USA). Finally, the obtained data were analyzed using FlowJo software (BD Life Sciences).

### 2.6. Cell Proliferation Analysis

The effects of the transgenes on BM-rMSC proliferation and viability were tested through WST-8 cell proliferation and trypan blue exclusion assays. The WST-8 cell proliferation assay (ab228554; Abcam, Boston, MA, USA) was performed based on the manufacturer’s instructions. Cells were seeded in 96-well plates at the number of 5 × 10^3^ cells/well. Each cell was assayed in triplicate. The cells were then maintained for 24 h (day 1), 48 h (day 2), and 72 h (day 3); subsequently, 10 µL of WST-8 reagent was added to each well. The reaction proceeded for 1.5 h at 37 °C under 5% CO_2_. The absorbance value of each sample at 470 nm was detected using a fluorescence plate reader. BM-rMSCs were seeded into 96-well culture plates at 10^5^ cells/well in triplicate and then incubated at 37 °C under 5% CO_2_ for the trypan blue exclusion assay. After 1, 3 and 5 days, the cells were stained with a 0.4% trypan blue solution. The viable cells were enumerated using an *EVE* automatic cell counter (NanoEnTek, Seoul, Korea) (Appendix A) with a hemocytometer.

### 2.7. Cell Cycle Progression

After the BM-rMSCs were harvested through trypsinization, they were washed twice by PBS and used 70% ethanol for fixation at −20 °C for at least 1 day. The fixed cells were then washed with ice-cold PBS and stained with propidium iodide solution (Biolegend) in the presence of 100 µg/mL RNase A (Sigma) for 30 min under dark conditions. The cell cycle progression was analyzed using a flow cytometer (BD Biosciences) and Multicycle AV software (De Novo Software, Glendale, CA, USA).

### 2.8. Western Blotting

The cells were lysed in a radioimmunoprecipitation assay buffer containing protease inhibitor cocktail (Roche, Mannheim, Germany). Pierce Coomassie Protein Assay (Bradford, Thermo Fisher Scientific, Waltham, MA, USA) was applied to measure the concentrations of protein extracts. Aliquots of protein extracts (40 µg/lane) were subjected to sodium dodecyl sulfate–polyacrylamide gel electrophoresis and transferred to phosphatidylcholine membranes (Sartorius, Göttingen, Germany). The membranes were blocked using skim milk and separately incubated overnight at 4 °C with primary antibodies against RFP (1:1000, MA5-15257, Thermo Fisher Scientific), Cre (1:1000, ab188568, Abcam), CDK4 (1:1000, MA5-12984, Thermo Fisher Scientific), cyclin D1 (1:200, MA5-16356, Thermo Fisher Scientific), cyclin A2 (1:500, 18202-1-AP, Proteintech), cyclin B1 (1:500, 55004-1-AP, Proteintech), and β-actin (1:5000, MA5-15739, Thermo Fisher Scientific). Subsequently, the membranes were probed using HRP-conjugated rabbit/mouse anti-IgG for 1 h at room temperature. Finally, protein bands were detected through enhanced chemiluminescence (ECL; PerkinElmer Life Science, Hopkinton, MA, USA) using a UVP Biospectrum (UVP, LLC Upland, CA, USA).

### 2.9. Immunohistochemistry (IHC)

Organ tissues were collected from transgenic rats and sectioned in the same direction. The samples were fixed with 4% paraformaldehyde for 1 h at room temperature and incubated with different percentages of dehydration buffers and wax to prepare paraffin-embedded sections (thickness of 6 µm). The tissue sections were deparaffinized in Sub-X Xylene Substitute (Leica, Richmond, CA, USA) and then incubated in Trilogy (Cell Marque, Rocklin, CA, USA) at 121 °C for 10 min for rehydration and antigen retrieval. Endogenous peroxidase was blocked using 0.3% hydrogen peroxide for 10 min. The tissue slides were washed with 1× Phosphate-Buffered Saline, 0.1% Tween 20 Detergent (PBST), and blocked with 5% BSA at room temperature for 1 h. The slides were then incubated overnight with antibodies at 4 °C. The slides were again washed with PBST, treated with the EnVision kit (Agilent Technologies, Santa Clara, CA, USA), and counterstained with hematoxylin and eosin. All the cover slides were observed through an Eclipse TE2000-U microscope (Nikon, Melville, NY, USA). Negative controls treated with only 5% BSA were also created.

### 2.10. Cell Differentiation Capacitiy

The capacity of BM-rMSCs to undergo adipogenic, osteogenic and chondrogenic differentiation was analyzed. BM-rMSCs from the third passage were used in the experiments. To analyze adipogenic and osteogenic differentiation, BM-rMSCs collected from the transgenic rats or obtained through RFP transduction were plated in a six-well culture plate at a density of 2.5 × 10^4^ cells/cm^2^. When the cells reached 40–50% confluence, adipogenic and osteogenic differentiation was initiated using StemXVivo Osteogenic/Adipogenic Base Media (CCM007, Bio-Techne, Minneapolis, MN, USA), with StemXVivo Adipogenic Supplement (CCM011, Bio-Techne) and StemXVivo Mouse/Rat Osteogenic Supplement (CCM009, Bio-Techne), respectively. The induction medium was replaced every 2–3 days, with caution taken not to disturb the cell monolayer. After 2–3 weeks, the cells were fixed with 4% paraformaldehyde for 30 min at room temperature (RT) and rinsed twice with PBS. To detect lipid droplet formation, histochemical staining was performed with 0.3% Oil Red O (Cod. O1391, Sigma-Aldrich). For analysis of osteogenic differentiation, calcium precipitates were analyzed with 40 mM Alizarin Red S (pH 4.1; Cod. TMS-008-C, Sigma-Aldrich). Subsequently, the cultures were washed three times and washed with PBS to remove nonspecifically bound stain. For induction of chondrogenic differentiation, fresh rMSCs were placed in a 15 mL centrifuge tube at a density of 5 × 10^4^ cells/cm^2^. The induction culture was prepared using StemXVivo Chondrogenic Base Media (CCM005, Bio-Techne) with StemXVivo Rat Chondrogenic Supplement (CCM020, Bio-Techne), which was replaced every 3 days for the cells to aggregate and form spherical cell pellets; these cell pellets were fixed, sectioned, and stained with Alcian Blue 8G (Cod. TMS-010-C, Sigma-Aldrich) to detect aggrecan, an indicator of cartilage formation. The morphology of the differentiated cells was imaged through inverted microscopy (Eclipse TS100; Nikon, Tokyo, Japan).

### 2.11. Semi-Quantitative Polymerase Chain Reaction

Differentiation was complete after 2–3 weeks, after which time the cells should exhibit the induced morphological changes. All the rMSCs were rinsed three times with PBS, and the attached cells were lysed with Trizol reagent (Thermo Fisher Scientific). After solubilization, chloroform was added, the supernatant (upper phase) containing the total RNA was transferred, and an equal volume of 100% ethanol was added. Subsequently, precipitation, washing, and elution were performed using Direct-zol RNA MiniPrep Kits (Zymo Research, CA, USA) according to the manufacturer’s protocol. The extracted total RNA was subjected to reverse transcription with SuperScript III reverse transcriptase (Invitrogen, Carlsbad, CA) to obtain complementary DNA. RT-PCR was performed using the Applied Biosystems 7900HT Fast Real-Time PCR System (Thermo Fisher Scientific). The cycling conditions were tested and further optimized for distinct sequences. PCR was performed for different numbers of cycles; expression levels were normalized to those of β-actin and analyzed using ImageJ v1.48 (National Institutes of Health, Bethesda, MA, USA). The forward and reverse primers used for genes examined are listed in Appendix A. The β-actin gene was used as an internal control in all PCR experiments.

### 2.12. Statistical Analyses

Data presented are the mean ± standard error of the mean (SEM) from at least three biological replicates. Analysis of variance (ANOVA) in GraphPad Prism 8 (GraphPad Software, San Diego, CA, USA) was used to identify statistical differences.

## 3. Results

### 3.1. Expression Analysis of DsRed and Cre from Transgenic Rats

To assess DsRed and Cre expression in organ tissues and BM-rMSCs, we performed IHC, RT-PCR, and Western blotting (Figure 1). Through IHC, DsRed and Cre expression was detected in several organs, including the brain, liver, kidney, and spleen. However, DsRed and Cre expression was nearly undetectable in wild-type (WT) tissue (Appendix A). Moreover, fluorescence microscopy was used to examine red fluorescence in the DsRed specimens, as illustrated in Appendix A, and in various organ tissues. Similarly, the DsRed and Cre mRNA and protein levels were expressed in both the organ tissue and BM-rMSCs. The mRNA and protein levels were further quantified, respectively, as shown in the panel below. Although the expression of transgenes in several organs had some differences, these results indicated the successful expression of DsRed and Cre in transgenic rats, enabling further investigation.

### 3.2. Identification of RFP Overexpression in BM-rMSCs

To elucidate the effects of the transgenes on differentiation capacity, we transduced *RFP* gene into BM-rMSCs using a lentivirus for comparison. After puromycin selection was performed for 14 days, RFP mRNA and protein levels were detected through RT-PCR and Western blotting, respectively. As illustrated in Appendix A, RFP overexpression was detected in the transfected cells but not in the control cells. Furthermore, the red fluorescence signal from the RFP-transduced cells was analyzed using both fluorescence microscopy and flow cytometry (Appendix A); both results indicated that RFP was successfully generated through lentiviral transduction.

### 3.3. Characterization of RFP-, DsRed- and Cre-BM-rMSCs

Transduction and genetic modification may lead to an imbalance of cellular functions. Stemness, a notable characteristic of MSCs, was evaluated in terms of the surface markers CD29, CD90, CD45, and CD11b/c. As the results of flow cytometry illustrate in Figure 2A, isotype control revealed extremely low levels (<0.7%) in control rMSCs, RFP-rMSCs, DsRed-rMSCs, and Cre-MSCs, indicating that the CD29, CD90, CD45, and CD11b/c stainings were reliable. In the control rMSCs, RFP-rMSCs, DsRed-rMSCs, and Cre-rMSCs, the percentages of the positive markers CD29 and CD90 were greater than 94.8%, whereas the percentages of the negative markers CD45 and CD11b/c were less than 0.8%. Furthermore, the immunofluorescence results (Figure 2B) indicated high CD29 and CD90 expression but the absence of the CD45 and CD11b/c surface markers. These data indicated that stemness was retained in the RFP-rMSCs, DsRed-rMSCs, and Cre-rMSCs, and no differences in the expression of surface markers were observed after transduction or genetic modification. Additionally, to elucidate the cause of the growth difference between the WT and transgenic rats (Appendix A), we analyzed cell growth by using the WST-8 assay, which analyzes cell proliferation on the basis of metabolic activity. The proliferation rate of distinct BM-rMSCs is illustrated in Figure 2C. The proliferation of the RFP-rMSCs, DsRed-rMSCs, and Cre-rMSCs obtained from the transgenic rats increased over time. However, the proliferation rates of the DsRed-rMSCs and Cre-rMSCs were remarkably lower than those of the control and RFP-rMSCs. The WST-8 assay revealed that the Cre-rMSCs had lower metabolic activity than the DsRed-rMSCs had after 1, 2 and 3 days. The growth differences (Appendix A) were corroborated by the results of the trypan blue exclusion assay (Figure 2D), which indicated cell viability. The viable numbers of control and RFP-rMSCs increased notably, whereas the viable numbers of DsRed-rMSCs and Cre-rMSCs did not. In summary, these results suggest that the DsRed-rMSCs and Cre-rMSCs had both compromised proliferation and elevated cell death.

### 3.4. Transgenes Hindered Cell Cycle Progression

To analyze the mechanism through which the transgenes contributed to the deceleration of cell growth, we performed a cell cycle assay. Flow cytometry (Figure 3A,B) revealed that, in the control rMSCs, 86.9% ± 0.6% of the cells were in the G0/G1 phase, 5.6% ± 0.8% were in the S phase, and 7.5% ± 1.3% were in the G2/M phase. However, the percentage of S phase cells in the DsRed-rMSCs and Cre-rMSCs was much higher. Although the percentage of DsRed-rMSCs and Cre-rMSCs in the G0/G1 phase decreased dramatically, the difference between the percentages of DsRed-rMSCs and Cre-rMSCs in the G2/M phase was negligible. Because the populations of these cells in the G0/G1 and S phases were lower and higher respectively than those of the control rMSCs, we assessed the changes in the levels of proteins encoded by cell cycle–related genes. As displayed in Figure 3C, the levels of cyclin D1, cyclin A2, cyclin B1, and CDK4 decreased in the DsRed-rMSCs and Cre-rMSCs. These results indicated that the transgenes hindered the cell cycle of the rMSCs, at least partly, by inhibiting the transition from the G1 to the S phase.

### 3.5. The Differentiation Effect of BM-rMSCs Collected from Transgenic Rats

MSCs are characterized by their unique ability to self-renew and to differentiate into multiple cell types, including adipocytes, osteocytes, and chondrocytes. We examined the capacity of these BM-rMSCs to differentiate into traditional lineages. After adipogenic, osteogenic, and chondrogenic induction in corresponding media for 21 days, the BM-rMSCs exhibited marked differences in morphology, as revealed by microscopic observation (Figure 4A–C). Moreover, the differentiated BM-rMSCs were stained by several reagents to identify the properties of different lineages. As illustrated in Figure 5A–C, evidence of adipogenic, osteogenic, and chondrogenic differentiation was observed after staining with Oil Red O for lipid droplets, Alizarin Red S for calcium deposits, and Alcian Blue for acidic polysaccharides, respectively. To validate these data, we further analyzed lineage-specific gene expression. Adipocyte protein 2 (aP2), a marker of adipocytes, was upregulated after adipogenic induction (Figure 5D,E). Similarly, dickkopf-related protein 1 (DKK1) was upregulated after osteogenic differentiation (Figure 5D,E). Type II collagen (formed by homotrimers of collagen, type II, alpha 1 chains), a specific marker of chondrocytes (Figure 5D,E), was also upregulated after chondrogenic induction. Thus, the results of lineage-specific gene expression and phenotypic analyses indicated no influence on cell differentiation capacity.

## 4. Discussion

In this study, RFP-rMSCs were created through the use of a lentiviral vector, which enables integration into the host cell genome. After several days of puromycin selection with RFP in BM-rMSCs, these RFP-rMSCs exhibited no differences from nontransfected BM-rMSCs in cell morphology. RT-PCR revealed high exogenous mRNA production. Furthermore, Western blotting revealed the RFP protein level to be high. Red fluorescence was detected either through flow cytometry or microscopy. The RFP-rMSCs were used as controls for comparing the effects of gene modification. In addition, distinct organ tissues and BM-rMSCs obtained from the transgenic rats were analyzed to validate our transgenic model. Moreover, immunophenotyping through flow cytometry and microscopy revealed that the genetic modification with RFP, DsRed, and Cre did not alter the MSCs’ surface markers.

The results of the WST-8 and trypan blue exclusion assays indicated that the proliferation rates and viability of the DsRed-rMSCs and Cre-rMSCs were considerably lower than those of the WT rMSCs. These results were validated through a cell cycle assay and Western blot analysis. The cell cycle comprises the series of events resulting in the division of a cell into two daughter cells; these events include the duplication of DNA and organelles and the subsequent partitioning of the cytoplasm and other components into two daughter cells through cell division. The G0/G1 phase is the nonproliferative or resting phase, DNA replication occurs during the S phase, the cell is ready for division in the G2 phase, and mitosis occurs in the M phase. According to our data, the DsRed-rMSCs and Cre-rMSCs exhibited an extended S phase. At least two possible mechanisms participate in cell cycle progression delay: The first one is that a distinct set of proteins that relay a cell from one stage to the next need to be phosphorylated by kinase, and the second one is that a cascade of checkpoints that monitor completion of crucial events and delay progression to the next stage if necessary. On the other hand, cyclins require being tightly regulated during cell cycle progression [31,32]. For instance, cyclin D1 is one of the first cyclins produced in the cell cycle. Downregulation of cyclin D1 can retard the transition to the S phase; thus, cyclin D1 plays a rate-limiting role in G/S transition [33,34]. Thus, our observations imply an effect of transgenic expression on cell proliferation, consistent with previous results. Overall, the levels of cyclins at each time point are crucial, and alterations in cell cycle regulatory proteins influence cell cycle progression and, consequently, stem cell proliferation.

The BM-rMSCs collected from the transgenic rats could be induced to differentiate into adipocytes, osteocytes, or chondrocytes, as confirmed through staining with multiple reagents [35,36,37] and through RT-PCR, which revealed increased mRNA expression of adipogenic, osteogenic, and chondrogenic markers. Therefore, these results indicate that transgenic DsRed and Cre expression does not affect the adipogenic, osteogenic, or chondrogenic potential of MSCs.

The Cre/*loxP* system is extensively utilized for spatial or temporal regulation of gene function in rats and mice [38,39,40,41]. In vivo observation of cell fusion or transdifferentiation by using the Cre/*loxP* system has been performed and the Cre/*loxP* system is regarded as a useful way [42,43,44]. Here, we proposed transgenic rat models by using the Cre/*loxP* technology and DsRed/emGFP dual-color fluorescence to test the possibility of cell fusion and transdifferentiation after the transplantation of BM-rMSCs into organs. Fusion or transdifferentiation products can be detected through the IVIS (in vivo imaging system) spectrum. Furthermore, the IVIS spectrum enables the imaging of bioluminescence intensity throughout an animal, not only in regions of interest. Molecular imaging can be used to locate Cre-expressing rMSCs after delivery *in vivo*, thus enabling imaging at desirable time points. Nevertheless, the transgenic rats grew at a much slower rate than the WT rats, as indicated by age and weight data (Appendix A). This growth-inhibitory phenomenon was consistent with the studies which showed that growth inhibition occurred as the expression of Cre recombinase in mammalian cells. Although numerous studies have investigated the overexpression of pluripotent genes in MSCs [45,46,47,48,49], few studies have explored transgenic expression within transgenic models, particularly by using the Cre/*loxP* system. Therefore, we investigated BM-rMSCs obtained from transgenic rats to ensure the models were suitable for in vivo application.

This study has a few limitations. First, the RFP-transduced rMSCs we used cannot fully represent the effect on cell proliferation compared to transgenes with *loxP* sites. The use of the same cassette but with the removal of *loxP* sites in rMSCs needs to be further executed. Second, the evidence on disrupted cell cycle progression requires an advanced investigation. The determination of cell cycle length is a key element in elucidating the parameters of cell growth kinetics. Third, as data show in Figure 1, the transgene expression level in distinct organs had some difference. Therefore, the use of this Cre/*loxP*-based strategy and DsRed/emGFP dual-color fluorescence in conjunction with the IVIS spectrum is noteworthy that this system may limit the application for detecting fusion products. Furthermore, fusion products take time to generate signals. The *DsRed* gene and the stop codon should be removed to initiate the transcription and translation of the *GFP* gene. Otherwise, the aforementioned delay hinders the immediate detection of fusion. Thus, our findings may underrepresent the actual frequency of cell fusion.

## 5. Conclusions

The proliferation of BM-rMSCs obtained using the Cre/*loxP*-based system decreased, and cell cycle progression was hindered because of decreases in the levels of cell cycle–related regulators. This effect was consistent with the inhibition in growth of transgenic models. However, the stemness and differentiation ability of the BM-rMSCs of transgenic rats were unaffected; thus, our models can aid in the understanding of the behavior of MSCs after stem cell treatment.

## Figures and Tables

**Figure 1 cells-11-02769-f001:**
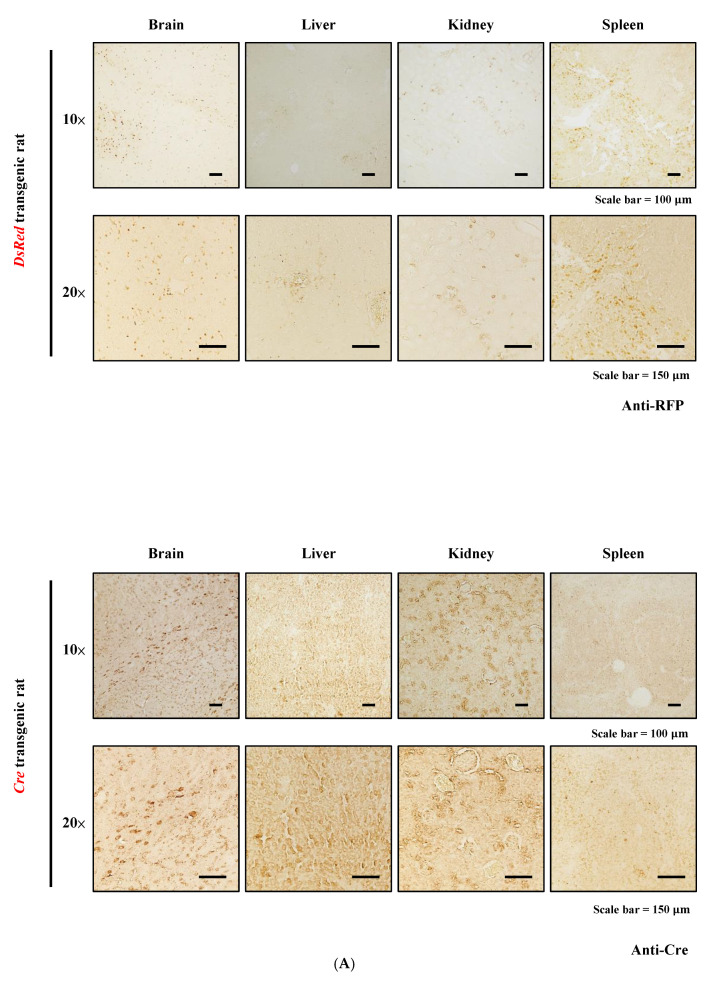
Transgene expression in organ tissues and BM-rMSCs. (**A**) IHC results for RFP and Cre antibodies. (**B**) RT-PCR results indicating transgene mRNA levels. Relative mRNA expression was further quantification (below panel). (**C**) Western blotting results for transgene protein levels in several organs and BM-rMSCs. Relative protein expression was further quantified (below panel).

**Figure 2 cells-11-02769-f002:**
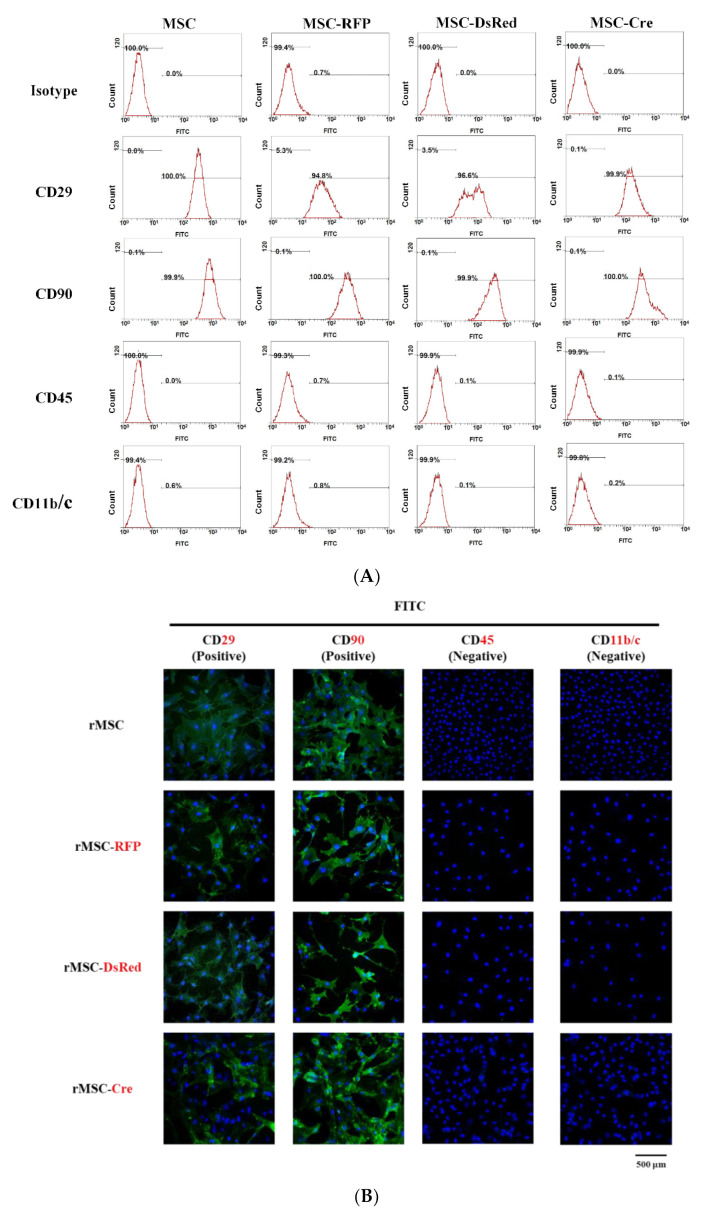
Stemness and proliferation assays of various rMSCs. (**A**) Control cells, RFP-rMSCs, DsRed-rMSCs, and Cre-rMSCs immunophenotyped for CD29, CD90, CD45, and CD11b/c through flow cytometry. The *x*-axis displays a single measurement parameter (relative fluorescence intensity). The *y*-axis displays the number of events (cell count). (**B**) Immunofluorescence of cells with different surface markers (green); nuclei were stained with DAPI (blue). (**C**) WST-8 proliferation assay. (**D**) Viable cell numbers counted through trypan blue exclusion assay. Data are representative of three independent experiments with similar results. Error bar represents SEM. The confidence interval was 95% considering the multiple comparisons between groups: ** *p* < 0.01 and *** *p* < 0.001.

**Figure 3 cells-11-02769-f003:**
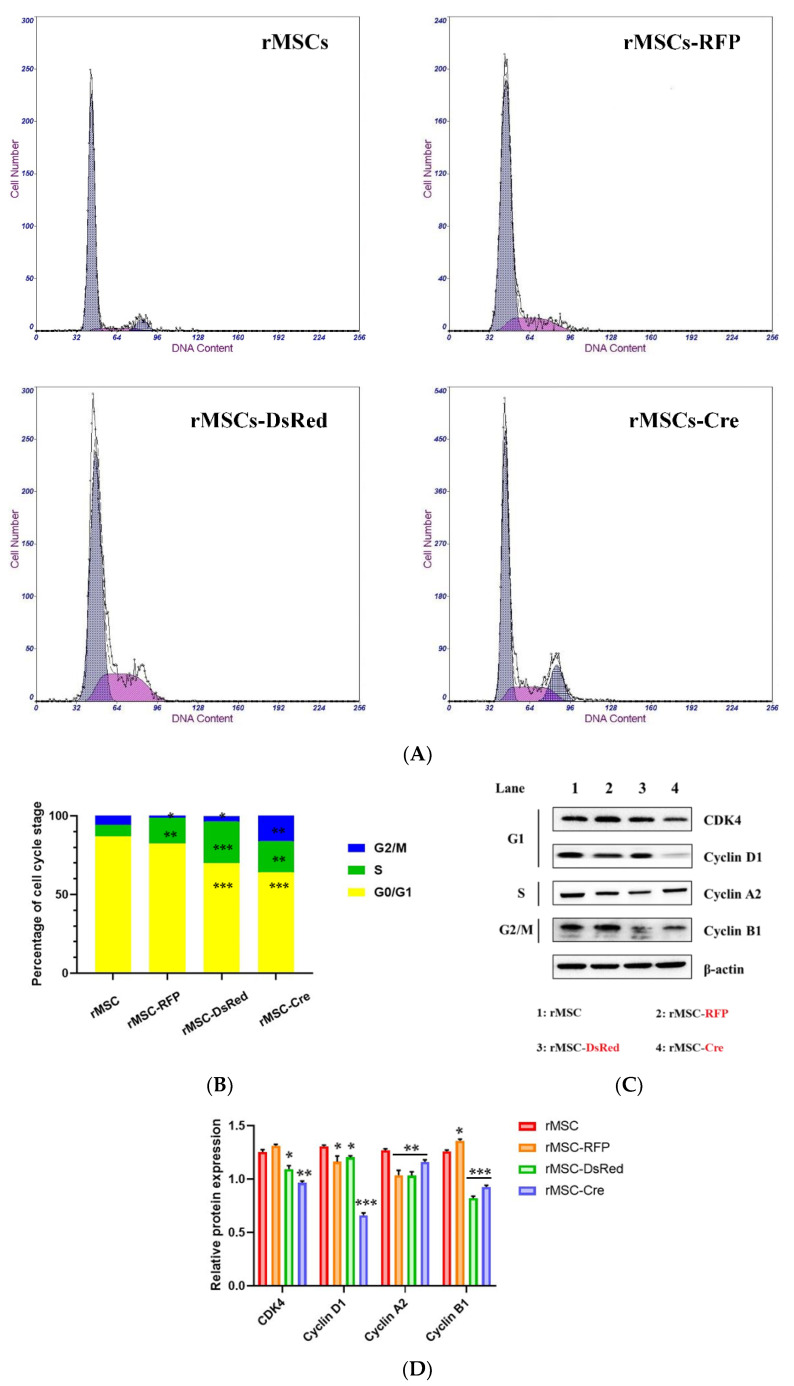
Cell cycle analysis in DsRed- and Cre-BM-derived rMSCs. (**A**) Representative flow cytometry results of control, RFP-, DsRed-, and Cre-BM-derived rMSCs. (**B**) Cell populations in each phase were quantified using Multicycle software. (**C**) Expression levels of major regulators within cell cycle using Western blotting. (**D**) Relative protein expression level was further quantification. Experiments were performed in triplicate, and data were analyzed using analysis of variance (ANOVA). Differences were considered significant at * *p* < 0.05, ** *p* < 0.01, *** *p* < 0.001.

**Figure 4 cells-11-02769-f004:**
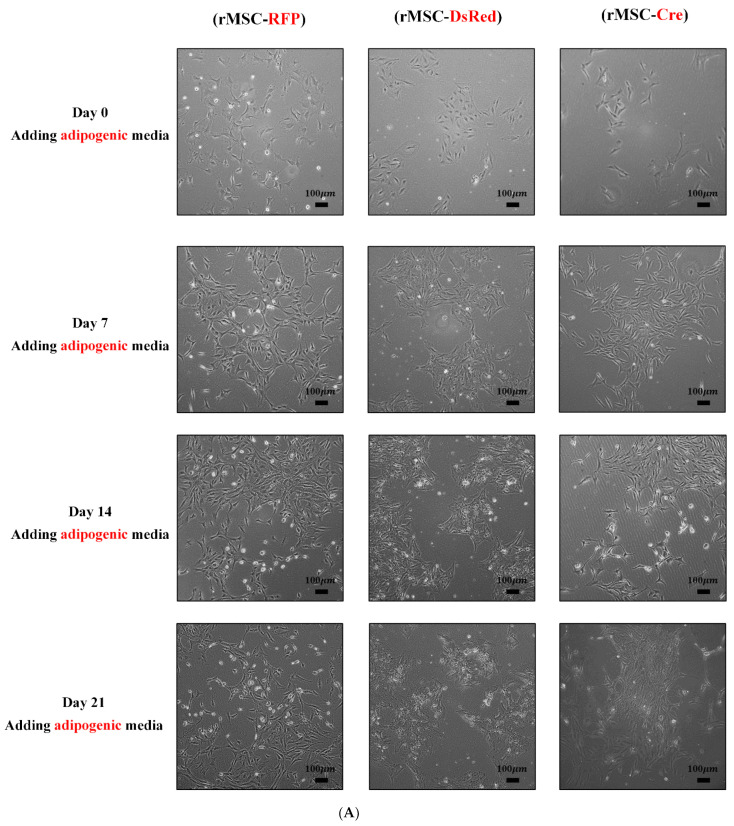
Morphology of differentiated rMSCs. BM-rMSCs displayed traditional lineages in vitro. All rMSCs were seeded in appropriate media for inducing differentiation. (**A**) rMSCs in adipogenesis-inducing medium formed varying sizes of lipid droplets. (**B**) Maximum calcium deposition produced by rMSCs grown in osteogenesis-inducing medium. (**C**) Chondrocyte pellet (ball) formed by rMSCs cultured with chondrogenesis-inducing medium.

**Figure 5 cells-11-02769-f005:**
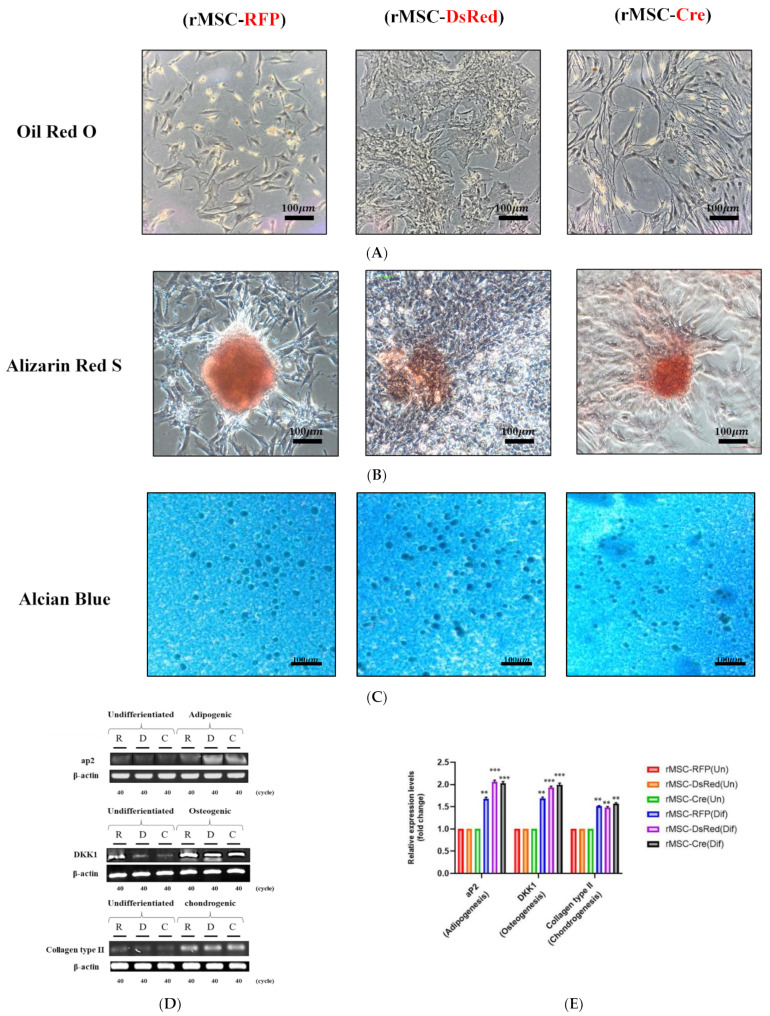
Results of histochemical staining and RT-PCR analysis of differentiated rMSCs. (**A**) Adipocytes stained with Oil Red O to detect lipid droplet formation. (**B**) Osteocytes stained with Alizarin Red S to detect calcium precipitates. (**C**) Chondrocytes stained with Alcian Blue to detect aggrecan. (**D**) Gene-specific primers used to detect mRNA. (**E**) Fold changes in mRNA expression of differentiated cells, evaluated semiquantitatively. Scale bars represent 100 μm. Data are representative of three independent experiments and were compared through analysis of variance (ANOVA). Differences were considered significant at ** *p* < 0.01, and *** *p* < 0.001.

## Data Availability

The data presented in this study are available on request from the corresponding author.

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
