# Peer review of "Differentiation Capacity of Bone Marrow-Derived Rat Mesenchymal Stem Cells from DsRed and Cre Transgenic Cre/loxP Models"

_cells, 2022, doi:10.3390/cells11172769_

Round 1
Reviewer 1 Report
Thank you for allowing me to review this manuscript to understand the proliferation, cell cycle progression, and differentiation of mesenchymal stem cells (MSC) in transgenic mice bearing Cre gene and loxP sequences. The authors have proved the cell cycle progression delay in the transgenic mice compared to that of control animals' MSC. However, there are no differences in the differential capacities of MSC collected from transgenic and wild-type mice. The authors have done well-established techniques and methodologies to prove the proliferation, cell cycle, and differentiation capacity of MSCs. However, there are major investigations or data that need to be included.
1. Investigators have based their hypothesis on the loxP and Cre technology, however, none of the data belongs to mice that bear no loxp after cre-mediated recombination. Investigators should include one more cohort where emGFP is expressed following Cre-mediated recombination.
2. Genetyping will also need to be included after Cre-mediated recombination.
3. Placement of Figures in the text (supplemental figures) is confusing. I could not find supplemental Figures.
4. In the discussion, investigators should indicate the possible mechanisms of cell cycle progression delay in transgenic mice. Or include some investigations to determine the possible mechanisms.
Author Response
Point 1: Investigators have based their hypothesis on the loxP and Cre technology, however, none of the data belongs to mice that bear no loxp after cre-mediated recombination. Investigators should include one more cohort where emGFP is expressed following Cre-mediated recombination.
Point 2: Genotyping will also need to be included after Cre-mediated recombination.
Response 1&2: Thank you for your suggestion. In this study, our purpose focus on examining whether any effects of BM-MSCs obtained from transgenic rats occurred owing to the growth difference. Thus, we did not include the genotyping data after Cre-mediated recombination. However, we have the preliminary data by using the IVIS spectrum to prove that our model really works and we successfully detected the emGFP signal after Cre-mediated recombination. In the future, we will confirm the genotyping after Cre-mediated recombination and keep on elucidating the behavior of MSCs after cell transplantation based on the models.
Point 3: The placement of Figures in the text (supplemental figures) is confusing. I could not find supplemental Figures.
Response 3: Sorry for the technical issue which causes the problem of finding supplemental data. We have uploaded the file. Please download the attachment.
Point 4: In the discussion, investigators should indicate the possible mechanisms of cell cycle progression delay in transgenic mice. Or include some investigations to determine the possible mechanisms.
Response 4: Thank you for your suggestion. In the discussion, we have indicated the possible mechanisms of cell cycle progression delay in the transgenic rats as below:
Line 421-425: At least two possible mechanisms participate in cell cycle progression delay: The first one is that a distinct set of proteins that relay a cell from one stage to the next need to be phosphorylated by kinase and the second one is that a cascade of checkpoints that monitor completion of crucial events and delay progression to the next stage if necessary.
Reviewer 2 Report
The study by Tseng et al. investigates the effects of the Cre/loxP transgenesis on cell proliferation and differentiation of rat bone marrow-derived mesenchymal stem cells (MSCs) using two transgenic rat models: SD-Tg(UBC-DsRedT3-emGFP)18Narl and SD-Tg(UBC-cre/ERT2)7Narl. They find that, whereas MSCs derived from the transgenic rats can differentiate normally into adipocytes, osteocytes, and chondrocytes, they show different properties in terms of cell proliferation and cell cycle progression. This is potentially an interesting study for uplifting awareness of these effects while using the Cre/LoxP system for cell tracking. Nonetheless, several aspects of the paper need to be improved (detailed below):
1. Probably due to technical problems, I cannot find some of the data (Table 1, 2 and 3) that they mentioned in the combined manuscript.
2. The rational of this study is not well explained. For example, in the first paragraph of the introduction, they mentioned the current lack of consensus on cell fusion and trans differentiation after MSC transplantation (Line 46-52). They also mentioned lack of techniques for cell tracking and talked about these two aspects again in the 4th paragraph of the discussion (line 361-377). All these descriptions make it sound like this study is developing a new imaging system to study cell fusion and trans differentiation after MSC transplantation. Especially, they specifically mentioned in line 62-63: “In the present study, we used this Cre/loxP-based molecular approach to detect fusion of transplanted cells to cells of organs of living rats.” However, they did not show any transplantation and cell fusion results.
3. They do not provide enough information about the research background of this study. For example, there are previous studies about the effects of Cre on cell death. They should mention these studies in their introduction and discussion.
4. For the SD-Tg(UBC-DsRedT3-emGFP)18Narl derived MSCs, since they use lentiviral transfected RFP transgenic cells as control, it is difficult to say if the effect on cell proliferation is due to the presence of the loxP sites. If they use the same cassette but remove the loxP sites, will they get the same results?
5. As mentioned in point 3, there are studies showing that Cre can cause DNA damage and cell death in the absence of loxP sites, it is therefore important for the authors to show whether or not cell proliferation will be affected if they combine the Cre and loxP by crossing their two transgenic strains.
6. In the Abstract, the authors should briefly explain their system first before talking about Ds-Red and Cre-rMSC.
7. Abstract, line 23, spell out the acronym “BM-rMSCs” on first mention.
8. Line 143, add source for WST-8 (company, cat.no. etc)
9. Figure 1 A, the resolution of the images needs to be improved.
10. For all their western blots and semi-qPCRs, the authors need to quantify and show their quantification results. If possible, they should confirm their semi-qPCRs with real time quantitative PCR.
11. Figure 2A, explain in the legend the meaning of the X, Y axis and the guidelines. Besides, in all these plots, the transgenic groups: MSC-RFP, MSC-DsRed and MSC-Cre seem more similar to each other than to the WT control?
12. Figure 2B, explain in the legend the Negtive images.
13. Figure 2C and 2D, the authors should use a statistical test for multiply group comparison (e.g. ANOVA) rather than t-test;
14. For cell cycle analysis, it is also important to measure cell cycle length in different conditions, especially if they think cell cycle progression is affected.
15. Figure 4C, it is not clear what we are supposed to see.
16. Figure 5C, Alcian Blue staining. The tissue is too thick to see tell if the staining is positive. They should try sectioning or use microMass culture to show real blue color from their staining.
17. Figure 5D, the authors should check type I collagen for osteogenesis.
18. Figure 5E, error bars are missing. Multiple group statistical comparisons are more appropriate.
Author Response
Point 1: Probably due to technical problems, I cannot find some of the data (Tables 1, 2, and 3) that they mentioned in the combined manuscript.
Response 1: Yes, maybe the technical problems so that you cannot find some of the data. We have uploaded the file and please download the attachment.
Point 2: The rationale of this study is not well explained. For example, in the first paragraph of the introduction, they mentioned the current lack of consensus on cell fusion and transdifferentiation after MSC transplantation (Line 46-52). They also mentioned the lack of techniques for cell tracking and talked about these two aspects again in the 4th paragraph of the discussion (lines 361-377). All these descriptions make it sound like this study is developing a new imaging system to study cell fusion and transdifferentiation after MSC transplantation. Especially, they specifically mentioned in lines 62-63: “In the present study, we used this Cre/loxP-based molecular approach to detect fusion of transplanted cells to cells of organs of living rats.” However, they did not show any transplantation and cell fusion results.
Point 3: They do not provide enough information about the research background of this study. For example, there are previous studies about the effects of Cre on cell death. They should mention these studies in their introduction and discussion.
Response 2&3: Thank you for your suggestion. We have rewritten the introduction and discussion and have provided enough information to make the rationale of this study clearer. Indeed, we tried to develop a new imaging system to study cell fusion and trans-differentiation after MSC stem cell treatment. However, we discovered a growth difference between wild-type rats and transgenic rats as shown in Table S2. Therefore, our purpose focus on examining whether any effects of MSCs obtained from transgenic rats occurred. In the future, we will keep on conducting experiments with the use of Cre/loxP-based molecular approach to detect the fusion of transplanted cells to cells of organs of living rats.
Point 4: For the SD-Tg(UBC-DsRedT3-emGFP)18Narl derived MSCs since they use lentiviral transfected RFP transgenic cells as control, it is difficult to say if the effect on cell proliferation is due to the presence of the loxP sites. If they use the same cassette but remove the loxP sites, will they get the same results?
Response 4: Thank you for your suggestion. Indeed, the RFP-transduced cells we used cannot fully represent the effect on cell proliferation compared to transgene with loxP sites. The use of the same cassette but with the removal of loxP sites needs to be further executed. This is our limitation and we have written in the discussion.
Point 5: As mentioned in point 3, there are studies showing that Cre can cause DNA damage and cell death in the absence of loxP sites, it is therefore important for the authors to show whether or not cell proliferation will be affected if they combine the Cre and loxP by crossing their two transgenic strains.
Response 5: Thank you for your suggestion. We have cited references showing that Cre recombinase has an impact on DNA damage and cell death in the absence of loxP sites. On the other hand, we will further study if the cell proliferation is affected or not where emGFP is expressed following Cre-mediated recombination.
Point 6: In the Abstract, the authors should briefly explain their system first before talking about Ds-Red and Cre-rMSC.
Response 6: Thank you for your suggestion. We have edited our manuscript and explained our system first before talking DsRed and Cre-rMSC. The schematic diagram of our models was shown in Table S1.
Point 7: Abstract, line 23, spell out the acronym “BM-rMSCs” on the first mention.
Response 7: Thank you for your suggestion. We have spelled out the acronym “BM-rMSCs” at the first mention. BM-rMSCs is an acronym for bone marrow-derived mesenchymal stem cells.
Point 8: Line 143, add the source for WST-8 (company, cat.no. etc)
Response 8: Thank you for your reply. The source for WST-8 was purchased from Abcam (ab228554). We have added the source in the manuscript.
Point 9: Figure 1 A, the resolution of the images needs to be improved.
Response 9: Thank you for your suggestion. We have provided better resolution images.
Point 10: For all their western blots and semi-qPCRs, the authors need to quantify and show their quantification results. If possible, they should confirm their semi-qPCRs with real-time quantitative PCR.
Response 10: Thank you for your suggestion. We have quantified the results of Western blots and semi-qPCRs as shown in Figures 1 B, C (below panel), Figure 3D, and Figure 5E.
Point 11: Figure 2A, explain in the legend the meaning of the X, and Y axis and the guidelines. Besides, in all these plots, the transgenic groups: MSC-RFP, MSC-DsRed, and MSC-Cre seem more similar to each other than to the WT control?
Response 11: Thank you for your suggestion. We have explained in the legend the meaning of the X, and Y axis and the guidelines as below:
The x-axis displays a single measurement parameter (relative fluorescence intensity). The y-axis displays the number of events (cell count).
MSCs express many surface markers such as CD29 and CD90. We then used positive markers to ensure that MSCs remain the stemness.
Figures 2A and B were used to illustrate that the stemness (transgenic groups versus control) has no difference. All the MSCs express positive markers (CD29 and CD90), while negative markers (CD45 and CD11b/c) do not express.
Point 12: Figure 2B, explain in the legend of the Negative images.
Response 12: Thank you for your suggestion. We have explained in the legend of the negative. It means that the CD markers do not express and could serve as a negative marker.
Point 13: In figures 2C and 2D, the authors should use a statistical test for multiply group comparison (e.g. ANOVA) rather than a t-test.
Response 13: Thank you for your suggestion. We have used the appropriate analysis for multiple group statistical comparisons.
Point 14: For cell cycle analysis, it is also important to measure cell cycle length in different conditions, especially if they think cell cycle progression is affected.
Response 14: Thank you for your suggestion. We have mentioned the importance of cell cycle length for examining cell cycle progression and written in the discussion. Although the measurement of cell cycle length is important, we think that our results from PI staining which indicate the percentage of each phase can be evidence for examining if the cell cycle is affected or not. In addition, western blot analysis showed that cell cycle-related proteins were downregulated.
Point 15: Figure 4C, it is not clear what we are supposed to see.
Response 15: Thank you for your suggestion. We have provided high-resolution photos and used arrow pointing to demonstrate the location of chondrocytes clearly.
Point 16: Figure 5C, Alcian Blue staining. The tissue is too thick to see tell if the staining is positive. They should try sectioning or use microMass culture to show real blue color from their staining.
Response 16: Thank you for your suggestion. We have tried sectioning to show real blue color after Alcian Blue staining. As figures are shown in Figure 5C.
Point 17: Figure 5D, the authors should check type I collagen for osteogenesis.
Response 17: Thank you for your reply. We hope to follow your suggestion and check type I collagen as an osteogenic marker. However, we could not repeat the osteogenesis right now since the differentiated media which we used will no longer be sold from the company and the shipping time for other brands also take time due to coronavirus disease.
Point 18: Figure 5E, error bars are missing. Multiple group statistical comparisons are more appropriate.
Response 18: Thank you for your suggestion. We have edited Figure 5E and added the error bars on it. Also, we used the appropriate analysis for multiple group statistical comparisons.
Round 2
Reviewer 1 Report
Thank you for adding new explanations and citations
Author Response
Thank you for your comments and recommendation. We re-summit our revised manuscript according to reviewer 2's suggestions and comply with the iThenticate Report Check requirements.
Reviewer 2 Report
The manuscript has been improved after the revision. With the help of the supplementary data, the rationale of the study becomes clearer. Although some of my points (point 4 and 5) were not experimentally addressed, the authors properly mentioned the limitations of this study, which seem fine to me. Below are some points that can be improved.
1. For Figure 1B and 1C, did the authors do statistical analysis and find no difference between different tissues? This seems to be the case based on their quantification results. The authors should mention it in the text.
2. There is inconsistency between their IHC (Figure 1A) and expression quantification results (Figure 1B and C). For example, the staining of DsRed expression in kidney and Cre expression in Spleen seem to be very faint. But there seem to be no statistical difference between tissue samples (point 1) in mRNA and protein expression. Also, Cre mRNA expression and protein expression do not seem to agree with each other based on the gel images. How many replicates did the authors use for quantification? In Figure S2, why is there positive staining in WT kidney for Cre?
3. Line 301-308, the trypan blue exclusion assay suggests more cell death rather than less proliferation. It’s likely that DsRed-rMSCs and Cre-rMSCs have both compromised proliferation and elevated cell death. The authors may need to incorporate this in their description.
4. Figure S3, are all the RFP signals supposed to be positive? Based on their images, the red signals for lung, liver and intestine are hard to tell. For DsRed, the excitation wavelength is 558nm not 488nm (which is for GFP). Is this just a typo or did they use wrong light source during imaging?
5. Table S2, how many animals were used in each group? Can they explain the error bars and why only some points have them?
Author Response
Point 1: For Figure 1B and 1C, did the authors do statistical analysis and find no difference between different tissues? This seems to be the case based on their quantification results. The authors should mention it in the text.
Response 1: Thank you for your suggestion. We did not do statistical analysis for Figures 1B and 1C. The data didn’t incorporate the control (WT) for comparison because of consideration of the 3R’s principle of the animal. We can do the statistical analysis only if we euthanize more animals that don’t fit the 3R’s principle. In our original design, we wanted to make sure that the transgenic rats did express transgenes in several organs. However, there exist some differences in expression levels between these organs in transgenic animals (mentioned in Line 246-251). Thus, it is noteworthy that our models may have the limitation for detecting fusion products in the future, and we have written it in the discussion as well. (Line 463-466)
Point 2: There is an inconsistency between their IHC (Figure 1A) and expression quantification results (Figure 1B and C). For example, the staining of DsRed expression in the kidney and Cre expression in the Spleen seem to be very faint. But there seems to be no statistical difference between tissue samples (point 1) in mRNA and protein expression. Also, Cre mRNA expression and protein expression do not seem to agree with each other based on the gel images. How many replicates did the authors use for quantification? In Figure S2, why is there positive staining in the WT kidney for Cre?
Response 2: Thank you for your suggestion. We used three replicates for quantification. The reason for inconsistency between IHC and quantification results may come from the tissue sections we chose and which part of tissue specimens we conducted for RT-PCR and Western blot. In addition, sometimes even if there are many mRNAs, they will not necessarily behave as much as the protein predicted to behave.
In Figure S2, we think there is background staining in the WT kidney for Cre. Background staining can be caused by inappropriate antibody binding or by mistakes during the preparation of the tissue slide. Endogenous peroxidase and biotin are very high in tissues such as the liver and kidney (tissues with more red blood cells). It is necessary to reduce background staining by prolonging the inactivation time and increasing the concentration of the inactivator. We have reconducted the experiment and replaced the image as shown in Figure S2.
Point 3: Line 301-308, the trypan blue exclusion assay suggests more cell death rather than less proliferation. It’s likely that DsRed-rMSCs and Cre-rMSCs have both compromised proliferation and elevated cell death. The authors may need to incorporate this in their descriptions.
Response 3: Thank you for your recommendation. We agree with you, and we revised the manuscript as below:
Line 299-306: “However, the proliferation rates of the DsRed-rMSCs and Cre-rMSCs were remarkably lower than those of the control and RFP-rMSCs. The WST-8 assay revealed that the Cre-rMSCs had lower metabolic activity than the DsRed-rMSCs had after 1, 2, and 3 days. The growth differences (Table S2) were corroborated by the results of the trypan blue exclusion assay (Figure 2D), which indicated cell viability. The viable numbers of control and RFP-rMSCs increased notably, whereas the viable numbers of DsRed-rMSCs and Cre-rMSCs did not. In summary, these results suggest that the DsRed-rMSCs and Cre-rMSCs had both compromised proliferation and elevated cell death.”
Point 4: Figure S3, are all the RFP signals supposed to be positive? Based on their images, the red signals for lung, liver, and intestine are hard to tell. For DsRed, the excitation wavelength is 558nm, not 488nm (which is for GFP). Is this just a typo or did they use the wrong light source during imaging?
Response 4: Thank you for your suggestion. Yes, it is a typo. Here, we need to describe clearly that the red fluorescence of these organ sections was observed by the Nikon G-2A filter set. The information on this filter can be found on the website as follows (https://www.microscopyu.com/techniques/fluorescence/nikon-fluorescence-filter-sets/green-excitation-filter-sets/green-excitation-g-2a-longpass-emission). We have revised the legend in Figure S3. Indeed, it is hard to tell that the lung, liver, and intestine have the red signal since the red fluorescence intensity is quite low in these organs as we captured the images. In our opinion, the reason may due to the low expression level of DsRed.
Point 5: Table S2, how many animals were used in each group? Can they explain the error bars and why only some points have them?
Response 5: Thank you for your suggestion. Three animals per group were used in Table S2. We used GraphPad prism 8 to graph error bars. However, sometimes no error bar appears for certain points on XY graphs. The reason is that if the error bar would shorter than the size of the symbol, Prism simply won't draw it, even if the symbol is clear. We have adjusted the symbol size. Please see the revised version in Table S2.